# OpenReview forum: "Detecting Strategic Deception with Linear Probes"
_ICML.cc/2025/Conference — ICML 2025 poster_

### Official Review · Reviewer_6A7F · 2025-03-13

**Overall Recommendation:** 4

**Summary:**

The paper evaluates whether deceptive behaviour in LMs can be detected by linear probes on the activations. The paper includes extensive experiments and ablations with strong positive results.

**Claims And Evidence:**

The paper is clear about its limitations and scope, e.g.,

>Thus, our experiments do not attempt to prove that white-box monitors can achieve this ambitious goal, but only validate and benchmark performance in simpler settings.

(Writing nit: what is "this ambitious goal" here?)

Given the scope of the work, the results are compelling and quite strong. E.g., clear generalisation of probes to more complicated domains.

The paper performs lots of ablations and is refreshingly upfront about its limitations and failure modes (e.g., displaying interesting failures in table 2).

**Essential References Not Discussed:**

Deception can also be directly incentivised by training, e.g., to get good feedback from systematically mistaken evaluators, see (and cite)

https://arxiv.org/abs/2312.01350
https://arxiv.org/abs/2411.02306
https://arxiv.org/abs/2402.17747

(I usually think of this as "deceptive reward hacking")

**Experimental Designs Or Analyses:**

The metrics used to compare different methods are sound and the paper pushes itself to have practical application in real-life by investigating the 1% FPR metric.

**Methods And Evaluation Criteria:**

The paper uses many different benchmarks, showing strong generalisation. and comparison to control experiments. (And further ablations)

**Other Comments Or Suggestions:**

> Given this challenge, we focus on “white-box” techniques that analyse the model internals, rather than just examining outputs.

Well, yes, black-box evals are hard, especially when the model is deceptive, but a similar critique can be made against interpretability, whereby, the model obfuscates its internal reasoning. I think you should just note this, presumably it is more difficult (and beyond current models) to hide internal reasoning (unless its optimised against)

**Please write a short section summary/outline at the beginning of each section (e.g., sec 3 and 4).**

Minor note: the terminology "strategic deception" is probably good for you to use, as it makes the goal-directed aspect clear. But I think that, actually, all deception is "strategic", i.e., intentional (cf Ward https://arxiv.org/abs/2312.01350 ).

> the dataset in Azaria & Mitchell (2023)

Explain the data set / facts more

> Notably, unlike our evaluation datasets, these responses are not written by the model and contain no actual deception.

You could expand on these points, e.g., "not written by the model", could be a limitation at finding the probe for the model's internal deceptive "intent"

I wonder if the probes would pick up on a different thing if you tried to find "fine-tuned" intent, rather than "prompted" intent. I.e., if you could train probes in the same way, but contrasting with a bit of fine-tuning (for deception vs honesty) instead of prompting towards those personas.


"We adapt a setting from Benton et al.
(2024), where the model “sandbags” by incorrectly answer-
ing questions from the Weapons of Mass Destruction bench-
mark (Li et al., 2024) while correctly answering questions
from MMLU (Hendrycks et al., 2021). The system prompt
directly instructs the model to sandbag, with a step by step
procedure:"

Isn't that set-up from van der weij?

**Other Strengths And Weaknesses:**

It feels like the paper is written for quite a niche audience, maybe that's fine, but I think you could do more to make it more accessible. As an example "One guard against scheming would be to reliably detect strategic deception; this serves as an important component of an AI Control approach" --- some readers may not be familiar with the specific meaning if AI control here, nd it could be explained. Similar comment where you could explain the meaning of "sandbag".

You could explain probes better for non-familiar readers, e.g., at the start of 3.1 say "A probe is..."

please include a short summary of the paper at the end

**Questions For Authors:**

What's going on, in Fig 3, with the honest answers to the insider trading confirmation?

**Relation To Broader Scientific Literature:**

The paper is clearly up-to-date with the relevant work on deception. However, the related work section does not do a good job of placing the paper in the broader field (it acts more like a summary) --- it would be better to edit this section to explain how your work is situated and relevant to the different subareas you discuss. e.g., probes can be used to detect model orgs, but how exactly is your work different? Similar comment on the SAEs paragraph, how does it compare to your work?

**Theoretical Claims:**

n/a

---

### Official Review · Reviewer_XoXw · 2025-03-15

**Overall Recommendation:** 3

**Summary:**

The paper explores the use of logistic regression probes, trained on the residual stream activations of an LLM, to detect whether the LLM is executing deceptive behavior.

Their results demonstrate that such white-box deception probes can be trained using teacher forced (off-policy) data, and transfer to detecting on-policy deceptive behavior from LLMs.

Finally, they compare against a number of baseline techniques, including black-box chain of thought monitoring. These baselines perform fairly well, but the authors acknowledge this well.

**Claims And Evidence:**

The authors make the following broad claims:

1. White-box deception monitors will be important for detecting LLM deception in the future.
	1. This is not a central point of the paper, but rather the motivation that is argued in the introduction. In general I agree with the authors arguments, and find the motivation compelling.
2. White-box monitors can be used to detect deception in LLMs.
	1. This is the central claim of the paper that is backed up empirically. The authors train probes on off policy contrastive pairs of deceptive and non-deceptive examples. They then test the probes on out of distribution examples of models executing on-policy deceptive behavior. Figure 4 shows favorable ROC curves on these held out tasks. Figure 3 shows some issues however --- in the low FPR regime, their probes false positive rate appears quite high in some settings.

**Essential References Not Discussed:**

The related works appears complete to me.

**Experimental Designs Or Analyses:**

I split the experiments up by the figures / tables in which the data was presented.

Table 1: Describes how the ground truth data for deception or not is collected for the on-policy evaluations. Insider trading and sandbagging, the collection seems reasonable. For roleplaying, I think the gpt4o judge may introduce a non-trivial amount of label noise.

Figure 3, 4: Main results. The experimental design is reasonable. I am not sure what the dotted line on figure 4 is? Also I think providing the recall at 1%FPR explicitly would be useful.

Figure 6: These are the baseline experiments. More detail could be provided on these experiments (fine to reference the appendix more explicitly).

**Methods And Evaluation Criteria:**

The proposed method, of using logistic regression to probe model activations, is well born out in the related literature as a feasible technique (the novelty in this work comes from applying it to deception).

The evaluations seem reasonable also. They test against a number of held out datasets. Research in LLM deception is not well developed currently, so there are not a huge number of different evals that can be done --- the authors pick 3 (insider trading has 2 variants) which is reasonable.

**Other Comments Or Suggestions:**

It would be nice to link to Figure 6 more clearly when discussing baselines.

Also a discussion of the numerical baseline results would be good.

**Other Strengths And Weaknesses:**

Strengths:

- Originality: The paper is novel in that it is, to my knowledge, the first paper that applies white-box methods to detecting LLM deception.
- Clarity: The paper is very clearly written. Additionally the experimental methodology is clear and reasonable for demonstrating the claims of the paper.

Weaknesses:

- The main weakness of the paper comes in significance. While the paper shows that white box methods can be used to detect deceptive LLMs, they do not convincingly show that these methods are better than conventional black box approaches.

**Questions For Authors:**

Q1) Are there any additional experiments you can run the demonstrate the difference between white and black box methods. For example, testing out more black-box baselines (models, prompts)?

Q2) Can you provide a discussion of the numerical baseline results in Figure 6?


## Update after rebuttal

No rebuttal given by the authors to my specific review. After reading the other reviews and response, I still lean towards accepting.

**Relation To Broader Scientific Literature:**

The works sits at the intersection of two subfield --- AI control (or monitoring) and LLM probing. In general, most AI monitoring papers take a black-box approach ([1], [2]), The novelty in this work comes from demonstrating that white-box approaches are feasible for detecting deception.

The work does not, however, comprehensively show that white-box methods are much stronger than black-box ones.

[1] - Greenblatt, Ryan, et al. "AI control: Improving safety despite intentional subversion." _arXiv preprint arXiv:2312.06942_(2023).
[2] - Baker, Bowen et al. "Detecting misbehavior in frontier reasoning models"

**Theoretical Claims:**

N/A

---

### Official Review · Reviewer_8PzZ · 2025-03-15

**Overall Recommendation:** 2

**Summary:**

The paper studies the problem of strategic deception in AI models by training linear probes on datasets that elicit dishonesty in certain ways and check whether the probes generalize to more realistic examples of deception. The paper provides experimental validation that the probes can distinguish between honest and deceptive responses, but often fire more on honest responses than in the control prompts, leading to an increased false-positive rate. An analysis of what the probe might be learning is presented, and the method is compared favorably to other honesty detection approaches.

**Claims And Evidence:**

The claims are clear and well-supported. There is some speculation in the paper, but this is clear and not framed as a claim.

**Essential References Not Discussed:**

References are very well-done; there are no obvious missing references.

**Experimental Designs Or Analyses:**

Yes, the experimental designs are sound.

**Methods And Evaluation Criteria:**

Yeah, they more or less make sense for the method presented, but ultimately feel unsatisfactory and don't provide a particularly strong answer to what the probes are actually doing and why.

**Other Comments Or Suggestions:**

One thing I'm ultimately interested in with white-box probing is ways of studying how something like this might generalize to models that are actually superhuman, and scheming might be harder to detect. Is it possible to make a more explicit safety case here? That is, if I'm given a very strong model that I know nothing about, do the methods presented in this paper give me any sort of useful information about the model that might allow me to update in a certain direction about the model's underlying predisposition to scheming or ability to deceive?

**Other Strengths And Weaknesses:**

There have been lots of papers on probes; it would be nice if the paper made a more convincing argument about what exactly its main contribution is to the literature.

**Questions For Authors:**

1. Probes are trained off-policy, right? Like on the 3.2.1 dataset? How much better does training the probes on-policy do?
2. What is Figure 3? How do I interpret these graphs?

**Relation To Broader Scientific Literature:**

The paper's contributions are tightly related to broader research in strategic deception and probing of large language models.

**Theoretical Claims:**

No specific theoretical claims are made.

---

### Official Review · Reviewer_Bctj · 2025-03-17

**Overall Recommendation:** 3

**Summary:**

The paper evaluates whether linear probes can effectively detect strategic deception in language models by monitoring their activations. The authors train probes on simple datasets (instruction pairs and roleplaying scenarios) and test if they generalize to realistic deceptive behaviors like concealing insider trading and sandbagging on safety evaluations. Their probes achieve high performance (AUROCs 0.96-0.999) and can catch 95-99% of deceptive responses while maintaining a 1% false positive rate on control data. The authors conclude that while promising for monitoring, current performance is insufficient as a robust defense against sophisticated deception.

**Claims And Evidence:**

The claims are mostly well-supported by empirical evidence. The authors test their probes on multiple datasets, conduct ablation studies, and are very transparent about limitations.

The high performance metrics across diverse deception scenarios provide convincing evidence for the effectiveness of their approach.

However, the paper acknowledges that current models exhibit relatively blatant deception, so the results may not generalize to more subtle forms of deceptive behavior or more capable models. These are the settings that seem most important – and as such, I think the biggest weakness of the paper is to shy away from ambiguous settings, rather than trying to grapple with analyzing them despite the challenges.

**Essential References Not Discussed:**

I don't know of any essential references that were not discussed in the paper.

**Experimental Designs Or Analyses:**

The experimental designs seem generally sound. The authors take care to isolate the effects of different design choices by testing various probe architectures, training methods, and aggregation strategies. The worst-case approach of setting thresholds based on control data (which assumes ~0% of deceptive data in the real-world) seems good methodological practice.

**Methods And Evaluation Criteria:**

As far as I can tell, the methods and evaluation criteria are appropriate for the problem.

**Other Comments Or Suggestions:**

> "To fit our probe we flatten across samples and sequence positions"

I think this could be made a bit clearer. I was initially confused by this description.

Something that I think would be interesting to study is whether the probes generalize to being applied to models even after training that introduces deceptiveness, or whether probing works at all in settings in which models have been trained to be deceptive. My impression is that Wen 2024 does not specify whether their probes were trained using activations pre or post RLHF. Another work that seems particularly relevant is [Williams 2024](https://arxiv.org/abs/2411.02306), in which the authors find that very strong tendencies to perform strategic deception arises from RL training: I'd be curious to know whether probes trained on these models that have received further RL training would work.

It would be interesting to see whoever training on additional data that is about deception but is not deceptive could reduce the spurious correlations that you see and you study more in appendix A.1.

**Other Strengths And Weaknesses:**

Nothing not already stated above or below.

**Questions For Authors:**

N/A

**Relation To Broader Scientific Literature:**

Maybe i missed this in the paper, but i think making clearer how work differs from prior work on probes is important: did none of the prior works assess generalization of deceptiveness probes to new settings? If that were the case, that would be somewhat surprising, but I think this is important for assessing the novelty of this paper: my impression is that much of the novelty is about assessing the generalization of deceptiveness probes. In that sense, the title may be somewhat misleading, as other works cited have already attempted to detect deception with probes – this work just does so across different settings and with a deeper analysis.

**Theoretical Claims:**

N/A

---

### Official Review · Reviewer_2iyz · 2025-03-21

**Overall Recommendation:** 4

**Summary:**

The authors address the question of whether linear probes trained on simple deception datasets can generalise to more complex examples of deception. They train linear probes for Llama-3.3-70b-Instruct using a simple dataset of contrastive pairs. They then test generalization to four out-of-distribution deception datasets. They find that probes can achieve >95% test AUROC when classifying honest from deceptive responses. Lastly, the authors analyze failure modes and offer speculation on how future probing methods might improve on these metrics.

## Update after review

The authors addressed my questions and concerns. I will maintain my recommendation of 4.

**Claims And Evidence:**

The empirical finding - that linear probes trained to detect deception in Llama-3.3-70b - is well-supported by evaluations on four different test datasets. They also compare to strong baselines including SAE activations and black-box methods. In the appendix, the authors also provide detailed explanations of their probe-fitting methodology.

As the authors only tested a single model, it is unclear whether this finding generalizes.

**Essential References Not Discussed:**

None that I know of

**Experimental Designs Or Analyses:**

Yes.

**Methods And Evaluation Criteria:**

Yes.

**Other Comments Or Suggestions:**

The relationship between harmfulness and deceptiveness is not investigated. It would be concerning if the probe catches 'benign' forms of deception but misses 'catastrophic' forms of deception. For example, in the Sandbagging eval which uses the WMDP dataset, each prompt could be evaluated in terms of its potential for harm, and the harmfulness scores could be correlated with whether the probe detects this successfully.

You could also consider doing multi-layer aggregation of activations a la Fort and Balakrishnan (2024). https://arxiv.org/abs/2408.05446

**Other Strengths And Weaknesses:**

As mentioned above it would be useful to have quantitative evaluations of probe generalization in more than a single model.

The authors select the deception threshold using 1% FPR. It is unclear whether the results are robust to a different choice of threshold, or why 1% is the correct threshold.

**Questions For Authors:**

Did you try training probes on the concatenated activations of all layers? That might eliminate the need to do a sweep.

**Relation To Broader Scientific Literature:**

Investigating probe generalization is significant for AI safety, as probes trained on existing data will typically need to generalise out of distribution in deployment. The finding here is an important update towards the effectiveness of white-box monitoring.

**Theoretical Claims:**

N/A

---

> ### Author Rebuttal · Authors · 2025-04-01
>
> Thank you for your detailed review!
>
> We agree that it would be useful to compare results on other models. During writing we ran experiments on gemma-2-27b and llama-3.1-70b and saw broadly similar results, but these results are not included in the main paper (and were not run with the latest datasets or all ablations).
>
> Similarly, we agree that training probes on activations concatenated across layers would be a useful experiment for future work to investigate. We did some preliminary tests of training probes at several layers and aggregating the results across each layer, but found it didn’t meaningfully improve performance so didn’t include the variation in the final paper.
>
> We agree the relationship between harmful and deceptive outputs is interesting. Harmfulness probes have been shown to work well (at least on the HarmBench dataset, e.g. https://arxiv.org/abs/2412.09565). Since probes are generally cheap to train and evaluate, we expect the best approach for real-world applications is to train a separate probe for harmfulness and deception, rather than hoping that a single probe transfers to both datasets. Another key difference is that LLM providers probably don’t want their model to ever generate harmful responses – while deceptive responses may be acceptable as long as the user is aware (or even desirable in certain contexts such as social deception games).
>
> We agree it could be helpful to sanity check correlation between harmfulness and deception, however. The example you give of only catching benign forms of deception would indeed be concerning (but seems unlikely).
>
> On the choice of 1% FPR, Figures 4 & 10 have ROC curves which show recall rates for different FPR. Overall we think it is necessary for probes to have a low false positive rate on control data, however the exact FPR to fix is certainly flexible. The primary reason for using 1% FPR in our analysis is to allow for easy comparison to previous works which also use this metric (Roger (2023) and Bailey et al. (2024)).

---

> > ### Comment · Reviewer_2iyz · 2025-04-01
> >
> > Thank you for the detailed comments. I will maintain my score of 4.

---

### Decision · Program_Chairs · 2025-05-01

**Decision:**

Accept (poster)

**Comment:**

Paper Summary: This paper investigates whether simple linear probes, trained on examples of deception, can generalize to more complex deceptive behaviours in LLMs. The paper specifically studies only Llama 70B Instruct model. The authors evaluate probes on diverse deception scenarios, such as insider trading and safety sandbagging, and demonstrate strong performance.

Reviewer summary: The paper received good reviews and all reviewers agree that this is a strong and relevant paper for safety and interpretability. Strengths include rigorous experiments and clear discussion of failure modes. Key concerns highlighted by reviewers include limited model testing (only evaluated on llama70b) and shallow comparison with black-box methods.

Recommendation: I recommend a weak accept.